# H-VISTA Immunohistochemistry Score Is Associated with Advanced Stages in Cutaneous and Ocular Melanoma

**DOI:** 10.3390/ijms25084335

**Published:** 2024-04-14

**Authors:** Andreea Cătălina Tinca, Andreea Raluca Szoke, Bianca Andreea Lazar, Emőke Andrea Szász, Alexandru Nicușor Tomuț, Adrian Horațiu Sabău, Iuliu-Gabriel Cocuz, Titiana-Cornelia Cotoi, Raluca Niculescu, Diana Maria Chiorean, Ioana Ancuța Ungureanu, Sabin Gligore Turdean, Ovidiu Simion Cotoi

**Affiliations:** 1Doctoral School of Medicine and Pharmacy, University of Medicine, Pharmacy, Sciences and Technology “George Emil Palade” of Targu Mures, 540142 Targu Mures, Romania; andreea-catalina.tinca@umfst.ro (A.C.T.); adrian-horatiu.sabau@umfst.ro (A.H.S.); raluca.niculescu@umfst.ro (R.N.); chioreandianamaria@yahoo.com (D.M.C.); 2Pathology Department, Mures Clinical County Hospital, 540011 Targu Mures, Romania; ohii.bianca@yahoo.com (B.A.L.); emoke.szasz@umfst.ro (E.A.S.); iuliu.cocuz@umfst.ro (I.-G.C.); sabin.turdean@umfst.ro (S.G.T.); ovidiu.cotoi@umfst.ro (O.S.C.); 3Pathophysiology Department, University of Medicine, Pharmacy, Sciences and Technology “George Emil Palade” of Targu Mures, 540142 Targu Mures, Romania; 4Faculty of Medicine, University of Medicine, Pharmacy, Sciences and Technology “George Emil Palade” of Targu Mures, 540142 Targu Mures, Romania; nicusortomut19@gmail.com (A.N.T.); hanca.ancuta@gmail.com (I.A.U.); 5Faculty of Pharmacy, University of Medicine, Pharmacy, Sciences and Technology “George Emil Palade” of Targu Mures, 540142 Targu Mures, Romania; titiana.cotoi@umfst.ro

**Keywords:** VISTA, melanoma, cutaneous, ocular, H-score, immunohistochemistry

## Abstract

Melanoma represents a public health issue. One of the biggest goals of current research is to develop new therapeutic options for patients affected by this aggressive tumor. We conducted a retrospective study including 105 patients diagnosed with cutaneous and ocular melanoma, with stages varying from pT1a to pT4b and pT4e, respectively, and we performed immunohistochemistry reactions with the new potential prognostic marker, VISTA (V-domain Ig suppressor of T cell activation). We quantified the expression by applying the H-score adapted for VISTA and divided the patients, based on the median value, into groups that presented high, low, and negative expression. Therefore, we obtained 65 cases with positive expression for cutaneous melanoma and 8 cases with positive expression for ocular melanoma. Forty-one cases presented high expression in cutaneous melanoma and three cases presented high expression in ocular melanoma. In cutaneous melanoma, analytic statistics showed that VISTA expression was associated with a high Breslow index, high mitotic count, high Ki67 expression, and advanced clinicopathological stage. The majority of ocular melanoma cases demonstrating a positive reaction were classified as stage pT3, whereas earlier stages showed a negative reaction. Our findings underscore a significant correlation between VISTA expression and key prognostic factors in melanoma. Looking ahead, the prospect of future randomized studies holds promise in corroborating the clinical relevance of our findings. By further elucidating the intricate relationship between VISTA expression and melanoma progression, new treatment strategies could be found, improving patient outcomes in this challenging neoplasm.

## 1. Introduction

Cancer and its complications represent a public health issue. A major goal of modern medicine is to develop better therapeutic options for patients. A revolutionary wave began with immunotherapy and culminated with the 2018 Nobel Prize (Drs. Allison and Honjo) for the discovery of T cell immune checkpoints, including CTLA-4 (Cytotoxic T-lymphocyte associated protein 4) and PD-1 (Programmed cell death protein 1). Subsequently, the immuno-oncology field has grown continuously [1,2].

Novel immunotherapies targeting the tumor microenvironment are being researched. One of these is the V-domain Ig suppressor of T cell activation (VISTA), also known as PD-1H. This is an inhibitory ligand, part of the B7 family, and homologous to PD-L1 (Programmed death-ligand 1). The ability of VISTA to work as both a ligand and a receptor has been demonstrated in mice and in vitro experiments. The major conclusion reached is that VISTA decreases cytokine production (IL-10, IFN-γ, and TNF-α) along with the suppression of T cells [3,4,5,6,7].

The marker is expressed in myeloid line cells (monocytes, macrophages, dendritic cells, and granulocytes) and has low expression in T cells (CD4+ and CD8+). Some studies also described expression in tumoral cells. Considering these results, further investigations and studies are being conducted on various types of cancer to develop a new therapy. Several studies have demonstrated the efficacy of anti-V-domain Ig suppressor of T cell activation (VISTA) treatment, particularly in the context of pancreatic cancer. These studies have investigated the potential of anti-VISTA therapy both as a standalone treatment and in combination with anti-programmed death-ligand 1 (PD-L1) agents. In individuals diagnosed with melanoma, particularly those with metastatic disease, resistance to PD1/PD-L1 therapy is commonly observed. In such cases, there is often an increased expression of VISTA in lymphocytes compared to samples collected prior to treatment. The expression of VISTA on T cells has been observed in both murine and human subjects, with notable effects primarily on CD4+ cells and lesser impact on CD8+ cells. This expression pattern is more commonly observed in patients undergoing immunotherapy or radiotherapy. It is important to mention that the expression of VISTA differs in each pathology [8,9,10,11].

Melanoma is one of the most aggressive tumors known. It can develop in the skin but also affects the mucous epithelia and eye. In the last decades, many treatments have been tested, from conventional chemotherapy to the development of anti-*BRAF* (B-Raf proto-oncogene, serine/threonine kinase) therapy to immunotherapy. The *BRAF* discovery was a game changer for the therapeutic management of patients. Around half of melanoma patients expressed *BRAF* and were candidates for targeted therapy, which improved their chances of survival. The most important morphological factors in melanoma are represented by the depth of invasion (Breslow index), Clark level (from I to V, describing the cutaneous layers infiltrated by the tumor), mitotic index, ulceration, and lymphatic and vascular invasions. The proliferation index Ki67 is also important for assessing cell proliferation and can provide important predictions for patients. The subtype of the tumor is also important, as superficial melanoma or lentigo malignant melanoma have a better prognosis compared with other types, such as nodular melanoma, mucosal melanoma, or acral melanoma [12,13,14,15].

Ocular melanoma, the second most prevalent form of melanoma following the cutaneous variant, presents a formidable challenge due to its heightened morbidity and mortality, particularly in advanced stages. Diverging from its cutaneous counterpart, ocular melanoma, particularly with uveal involvement, exhibits a distinctive propensity for dissemination through blood vessels, with the liver emerging as its primary site of metastasis. One notable prognostic factor involves the invasion of the optic nerve, highlighting the relevance of morphological assessment. Compared to cutaneous melanoma, uveal melanoma introduces distinct variables into the prognosis equation, including macrophage infiltration, intratumoral blood vessels, tumor dimensions (height, diameter, and diopter), and mitotic activity, each evaluated meticulously for a comprehensive understanding of disease progression [16,17].

VISTA expression in melanoma, both cutaneous and ocular, is little known. Patient prognoses are influenced by the morphological parameters mentioned above. So far, studies evaluating the immunohistochemistry expression of VISTA in patients with melanoma have concluded that its positivity is associated with poor survival, by contributing to immune suppression and evading immune surveillance; therefore, it negatively impacts the prognosis [18,19,20].

The spectrum of treatment options available for melanoma ranges from surgical interventions to radiation therapy. The surgical excision of melanoma is the primary therapeutic option in the patients affected, serving as an important tool for the diagnosis while also being curative for melanoma in situ. Depending on the tumor stage, other alternatives have been developed over time, with chemotherapy holding a very vast historical importance. For cutaneous melanoma, target therapy and immunotherapy are continuously developing. Radiotherapy is also an important tool. Combinations of chemotherapy and radiotherapy have been administrated either concomitant or sequential, to obtain better local results. Studies suggest that combining radiotherapy with immunotherapy is most effective when the tumor has developed resistance to the immune system. Data provided in the literature so far describe the improved survival rate in patients with combined therapy (for, i.e., ipilimumab and radiotherapy) for locally advanced stages (more than IIIb). For patients with metastatic disease, the results are contradictory, and clinical trials are ongoing [21,22].

Regarding uveal melanoma, enucleation followed by oncological therapy has been consistent over time. An important milestone was reached in 1985, with the development of interventional radiotherapy (IRT), which allows the patient to preserve visual acuity in most cases. The results observed in patients treated with IRT are significant tumor regression, decreased vascularity, and low risk for metastatic disease. The response of tumors is recognized to be heterogeneous, as some uveal melanomas will respond faster than others. Based on this response and the impact IRT has on the tumor height, four major types of patterns have been described: type M (miscellaneous), D (decrease), S (stable), and I (increase). In clinical practice so far, the most common type described was type D. The M type category showed a variety of situations, in which the response either remains stationary (with important shrinkage initially) or shows significant shrinkage after a while (maximum 60 months after IRT). Nevertheless, based on the findings and the impact on patients’ prognosis and quality of life, IRT is a valuable therapeutic option, and it should be recommended on a large scale. High-dose-rate interventional radiotherapy is an important procedure that can be used for other types of ocular melanoma, such as conjunctival melanoma [23,24].

## 2. Results

A total number of 65 cases out of 92 showed positive expression for VISTA in cutaneous melanoma, 56 of them for nodular melanoma and 9 for superficial spreading melanoma. For ocular melanoma, 8 out of 13 cases were positive, all corresponding to uveal melanomas.

The maximum score for our cases reached a value of 150 and the lowest was 10. Samples with an H-score of zero were considered negative. Based on these results, we divided the patients into two distinct groups, one with high VISTA score (Figure 1) and low VISTA score category respectively (Figure 2), with the cut-off between being the value of 80, according to the median value. 

Thus, 41 patients diagnosed with nodular melanoma presented a high H-score and 15 patients diagnosed with the same subtype presented a low H-score. In regards to superficial spreading melanoma, nine cases showed a positive expression highlighted by a low H-score (Table 1). 

In uveal melanoma, five cases presented a low H-score and three cases presented a high H-score (Table 2). The predominant low H score in these cases is exemplified in Figure 3.

The cells marked were identified as macrophages, monocytes, and lymphocytes, identified on the hematoxylin–eosin stain, and confirmed with CD68 and CD3 immunostaining. To identify the lymphocyte subtype, we used antibodies CD4 and CD8 for immunostaining.

Most cases with a high H-score of 31, corresponding to nodular melanoma, showed the predominance of CD8+ cells in the TME, while the other 10 showed a predominance of CD4+ cells. Superficial melanoma cases showed a predominance of CD4+ cells in all nine cases (Table 3).

Patients with uveal melanoma showing positive expression of VISTA exhibited the following features: one case classified as pT2a with a high H-score, two cases classified as pT3a with a high H-score, and five cases classified as pT3a with a low H-score.

The analytic statistics provided important data, as follows. Patients demonstrating positive expression of VISTA displayed statistically significant correlations with advanced stages at the time of initial diagnosis, greater Breslow thickness (>2 mm), a higher Ki-67 proliferation index (≥25), and an increased number of mitoses (all *p* < 0.001). Furthermore, tumors in patients with positive VISTA expression were notably more prone to ulceration, and the peritumoral stroma exhibited a moderate to high presence of inflammatory cells.

The comparison between high H-score and low H-score patients provided the following data: patients exhibiting high levels of VISTA expression were notably older in comparison to those with low expression levels (*p* < 0.001). The high VISTA expression group presented more advanced stages at the time of initial diagnosis, with significantly elevated occurrences of Breslow thickness > 2 mm and ulcerated melanoma (all *p* < 0.05).

Immunohistochemistry demonstrated a moderate to high abundance of macrophages and lymphocytes in the peritumoral stroma of patients exhibiting high VISTA levels, as evidenced by antibody staining. Moreover, the pagetoid spread was more commonly detected in patients with low VISTA expression (*p* = 0.004, OR: 0.1). Notably, there was a trend towards statistical significance regarding Clark levels 3 to 5, the presence of an in situ component, and mitotic count (*p*-values: 0.09, 0.06, 0.05, respectively) (Table 4 and Table 5).

The comprehensive statistical analysis, covering all three groups categorized by VISTA expression (high, low, and negative), unveiled significant disparities among age groups (*p* = 0.001), with patients demonstrating high VISTA expression being notably older in contrast to those with low and negative expressions. Additionally, patients with high VISTA expression exhibited a higher incidence of advanced stages, Breslow thickness > 2 mm, Ki-67 proliferation index ≥ 25, Clark levels 3 to 5, tumor ulceration, and an increased number of mitoses (all *p* < 0.05). Furthermore, a significantly elevated presence of macrophages and lymphocytes was observed in these patients (Table 6).

Conversely, none of the analyzed factors, including positive surgical margins, perineural/lymphovascular invasions, microsatellitosis, and metastasis, exhibited statistical significance in any of the performed analyses (all *p* > 0.05).

## 3. Discussion

Methods for interpreting and quantifying immunohistochemistry reactions with VISTA have long been controversial. The interpretation was frequently laborious and time-consuming, requiring several pathologists to ensure accurate reporting. The use of the H-score (for example, commonly used to quantify estrogen and progesterone receptor expression) has been implemented by some pathologists for VISTA. This score is calculated based on the following formula: 3 × HIGH + 2 × MODERATE + 1 × LOW, where HIGH is the percentage of marked cells showing maximum intensity, MODERATE is the percentage of marked cells showing moderate intensity, and LOW is the percentage of poorly marked cells. The interval of obtained values can thus range from 300 to 0 points. The maximum intensity calculated in our cases reached 150 (40 × 3 + 10 × 2 + 10 × 1), and the lowest was 10 (0 × 3 + 0 × 2 + 10 × 1 + 90 × 0). Samples with an H-score of 0 were considered negative. Based on the H-score, we divided the patient categories into high and low VISTA scores, with the boundary between the two given by the median range of 150–10, with a value of 80 [25].

In our study, VISTA expression was identified in inflammatory cells (lymphocytes and macrophages) in the tumor microenvironment and was identified by their morphological appearance in HE stains and by immunolabeling with CD68 and CD3. The resulting data was consistent with literature values. In some studies, VISTA has also been expressed in tumor cells, granulocytes, or lymphocytes; however, the marker is mainly positive in inflammatory cells of the myeloid line, both mature and immature. VISTA expression on tumor microenvironment (TME) macrophages is associated with an advanced stage and poor prognosis [26].

VISTA expression in the tumor microenvironment has been analyzed in certain types of cancers, including some sarcomas (e.g., fibrosarcoma), mucosal squamous cell carcinoma, pancreatic adenocarcinoma, hepatocellular carcinoma, prostate adenocarcinoma, mesothelioma, and melanoma. For mesothelioma, high VISTA expression was correlated with a better prognosis according to the Kaplan–Meier curve. Patient survival increased by 6% for every 10% increase in immunohistochemical expression of the marker [27].

In some cases, VISTA expression has been identified in inflammatory cells in the myeloid line of the tumor microenvironment and in tumor melanocyte cells and it has been correlated with a poorer prognosis in these situations. VISTA expression in tumor cells is generally associated with increased mortality, particularly with malignant melanocytic tumors [19,28,29]. In our study, patients with positive VISTA expression exhibited statistically significant associations with more advanced stages at the initial diagnosis, increased Breslow thicknesses, higher Ki67 proliferation indices, and an elevated number of mitoses. Additionally, tumors in patients with positive VISTA expression were significantly more frequently ulcerated, and the peritumoral stroma exhibited a moderate to high number of macrophages and lymphocytes, as indicated by immunohistochemistry using the CD68 antibody and CD3 antibody.

Patients with high VISTA expression levels were significantly older than those with low expression levels. Moreover, the high VISTA expression group exhibited advanced stages at initial diagnosis, with significantly higher frequencies of Breslow thickness > 2 mm and ulcerated melanoma. Immunohistochemistry revealed a moderate to high presence of inflammatory cells in the peritumoral stroma of patients with high VISTA levels, as indicated by IHC reactions. Additionally, pagetoid spread was more frequently observed in patients with low VISTA expression. Clark levels III to V, in situ components, and mitotic counts showed a trend toward statistical significance.

According to a study by Kuklinski [18], increased positivity of VISTA in inflammatory cells in melanoma is associated with an advanced tumor stage and implicitly with reduced survival, risk of recurrence, and higher risk for metastasis. Choi led a study including 136 skin melanomas. Analysis of VISTA expression was correlated with an advanced clinical-pathological picture. High immunolabel expression was associated with reduced survival, an increased Breslow index, and a clinically and pathologically advanced stage [20]. Another study mentions that although VISTA was positively associated with survival, according to the Cancer Genome Atlas Program database, the immunohistochemical expression of the marker is now associated with an advanced stage and increased mortality [6,25,28]. Regarding ocular melanoma, data have been published stating that tumor aggressiveness and low survival rates correlate with positive VISTA expression in inflammatory cells.

Published studies on VISTA expression in melanoma have so far focused on cutaneous melanoma; thus, the literature contains minimal data on ocular melanoma. Positive VISTA expression has been described in cases of uveal melanoma at the level of the tumor microenvironment. Our study included 13 cases: 12 uveal melanomas and 1 conjunctival melanoma. VISTA was negative in conjunctival melanoma and in two stages of uveal melanoma (pT1a). VISTA immunolabelling was expressed in eight cases of uveal melanoma, with low or high scores, in cells in the tumor microenvironment identified as macrophages and lymphocytes. One of these cases showed membranal and cytoplasmatic positivity in melanocytic tumor cells.

Ocular melanoma is the most common type of melanoma after its cutaneous counterpart. Most of the tumors affect the choroid and cause visual impairment. The treatment often includes enucleation followed by oncological therapy to avoid spread. A very important therapeutic method is highlighted by key findings from patients undergoing IRT. This treatment resulted in significant tumor regression, decreased vascularity, and a reduced risk of metastatic disease. It is important to acknowledge the varied response rates among uveal melanomas, with some tumors showing faster responses than others. For conjunctival melanoma, the literature describes two such cases treated with a target dose of 34 Gy over the course of 5 days. Both cases presented a great outcome, without important side effects or long-term consequences [23,29].

Regarding IHC investigations, the diagnostic markers for melanoma are mandatory, but the prognostic factors differ. Two important parameters cutaneous melanoma does not follow are the density of blood vessels and the number of macrophages. Some studies highlighted VISTA expression in the tumor microenvironment and associated it with a poor prognosis. The discussion is extremely complex, as researchers have studied the expression of this marker in various tumors. Studies targeting uveal melanoma also discuss the variety of data published so far. For example, in breast cancer, VISTA expression was associated with a better prognosis in triple-negative patients [29,30]. In other types of tumors, such as gliomas, endometrial cancer, prostate cancer, and mesothelioma, the expression varies after treatment; however, in most of these cases, positivity and high expression are related to an advanced stage and poor outcome [31,32,33,34].

The subtype of cells present in TME is also important. It is a known fact that most tumors present with a microenvironment that is hostile to the immune function of T cells. Usually, in all tissues, T helper cells are predominant by comparison with T cytotoxic cells that are present in lower numbers. The ratio between these cells is changed when pathologies, such as malignant tumors, are involved. For most solid cancers, the number of CD8+ cells (cytotoxic cells) is very important and their function in TME has been more studied compared with CD4+. The high presence of cytotoxic cells has been associated with better survival, while in melanoma, their abundance is linked with an overall better outcome in terms of recurrence risk and metastasis risk. Adoptive cell therapy with tumor-infiltrating lymphocytes has been studied and applied recently in patients diagnosed with metastatic melanoma and provided optimistic results. Researchers obtained encouraging outcomes even in patients who followed multiple therapeutic lines in the past, from anti-*BRAF* therapy to anti-PD1/PD-L1 [35,36,37,38]. Even more, tumor-infiltrating lymphocytes are a known independent predictor for survival and lymph node status in patients with melanoma [38]. In our observations, we have noticed a slight predominance of CD8+ cells, particularly in cases where there is an elevated expression of VISTA-abundant inflammatory cells. This trend is notably associated with nodular melanoma cases. Conversely, in cases of superficial spreading melanoma, the classic predominance of CD4+ cells persists, respectively, in cases with low H-score expression. These findings are noteworthy, considering the literature findings regarding CD8+ (overall better prognosis) and VISTA (overall worse prognosis). Their association with TME clearly calls for additional and more thorough studies in the immediate future. It is important to emphasize that cases lacking VISTA expression require further examination as well. Delving deeper into these instances is crucial for gaining a better understanding of their immune profile and the possible consequences for disease advancement or response to therapy.

## 4. Materials and Methods

We conducted a retrospective observational study by selecting 105 cases, of which 92 were represented by primary cutaneous melanoma (78 nodular melanomas, 12 superficial spreading melanomas, and 2 nevoid melanomas) and 13 by primary ocular melanoma (1 conjunctival melanoma and 12 uveal melanomas). All cases were diagnosed in the Pathology Department of the Clinical County Hospital Mureș, in Târgu Mures, Romania, within the timeframe spanning 2017 to 2022.

The inclusion criteria considered patients diagnosed with primary cutaneous melanoma stages pT1a–pT4b with brisk inflammatory infiltrate and primary ocular melanoma stages pT1a–pT4e. Considering the rare occurrence of ocular melanoma, all cases diagnosed in our hospital were included in the study. Patients excluded from the study encompass those with recurrent melanoma devoid of primary tumor documentation, those exhibiting metastases at first diagnosis, individuals diagnosed with other tumors or cutaneous pathologies, and those with melanoma in situ, without invasive features.

Tissue samples for laboratory diagnostics were collected and processed using a standard HP method. The samples were embedded in paraffin and preserved in 10% neutral buffered formalin before undergoing hematoxylin–eosin staining. Diagnostic immunohistochemistry (IHC) analysis was conducted with an automated immunostainer (Benchmark GX; Ventana Medical Systems, Inc.; Tucson, AZ, USA) (project identification code: 19059/2022, data we previously published) [39].

The primary objective of the present study was to analyze VISTA expression in the selected cases. To achieve this, we conducted immunohistochemistry (IHC) reactions for this marker, adhering to the following protocol: rehydration of paraffin-embedded sections, followed by antigen retrieval in a recovery solution (Leica Microsystems, Germany) with a pH of 9, utilizing microwave heating at 850 W for 25–30 min. Endogenous peroxidases were effectively blocked using 3% hydrogen peroxide (H_2_O_2_) for 10 min. The sections were rinsed in saline solution (TBS) thrice, with each rinse lasting 5 min. VISTA rabbit monoclonal [EPR21050] primary antibody (dilution of 1:1000 Abcam, Cambridge, UK) was applied and the sections were left to incubate overnight at 4 °C. Following incubation, the sections underwent another set of TBS rinses (3 × 5 min) (Dako, CA, USA) and were subsequently incubated for 1 h at room temperature with goat anti-rabbit IgG H&L (HRP) (1:500 dilution; Abcam, Cambridge, UK). Visualization of the staining was accomplished using 3,3′-diaminobenzidine (DAB), and the slides were ultimately counterstained with hematoxylin in accordance with the manufacturer’s instructions. Depigmentation was performed to avoid interpretation problems where abundant melanic pigment was present.

To evaluate VISTA IHC reactions, we used a scoring system called H-score, customized specifically for this antibody. Typically, the H-score quantifies marker expression in neoplasms such as breast cancer (used to calculate the expression for ER and PR). However, for VISTA, we adjusted it to focus on the tumor microenvironment (TME) considering the presence of the marker in inflammatory cells. The H-score considers both the intensity of the stain and the percentage of cells stained. The score is calculated using the following formula: we multiply the count of cells with the highest intensity by 3, the count of cells with moderate intensity by 2, and the count of cells with weak intensity by 1. The maximum achievable score is 300, while a score of 0 indicates a negative reaction.

The H-score was rigorously assessed by two pathologists independently to uphold objectivity and minimize potential biases. In cases where the discrepancy between calculated values exceeded 10, a third pathologist was consulted to provide additional evaluation, ensuring thoroughness and reliability of the analysis.

The inflammatory cells, delineated by the antibody stain, were meticulously examined using standard staining techniques, leading to their identification as macrophages and lymphocytes. To corroborate these observations, immunohistochemistry reactions targeting CD68 and CD3 were utilized, further validating the characterization of these cellular populations. To identify T cell proportion, we performed IHC reactions for CD4 and CD8, and the cell count was estimated by three individual pathologists.

For statistical analysis, continuous variables were presented as the median and interquartile range (IQR). Categorical data were summarized using frequencies and percentages. The chi-square and Fisher’s exact tests with Yates’s correction were used to compare categorical data, particularly where expected frequencies below 5 were found. The Mann–Whitney U-test and Kruskal–Wallis one-way analysis of variance tests were applied to compare continuous non-parametric variables. All tests were two-sided, with *p* < 0.05 considered statistically significant. Data analysis was conducted using GraphPad Prism version 8.0.0 software (GraphPad, CA, USA).

Regarding the morphological parameters highlighted in the statistical analysis, we defined “advanced stage” as at least stage pT3a (stage III represents the stage from which target therapy can be indicated). The Breslow cut-off was set at a minimal value of 2 mm. For the proliferation index, Ki67, we fixed the threshold at 25%, representing the median value observed throughout our cases. This selection offered an equitable portrayal of the proliferative activity among tumor cells in our dataset. Regarding the Clark level of invasion, we restricted our focus to stages III to V, as these stages signify the vertical growth phase of the tumor, providing crucial insights into its invasive potential and clinical behavior.

## 5. Conclusions

Our findings underscore a significant correlation between immunohistochemical VISTA expression and key prognostic factors in cutaneous melanoma. Notably, high VISTA expression aligns with an increased tumor stage, increased Breslow index, increased tumor proliferation index (ki67), and increased mitosis count, indicating a potential role of VISTA in disease aggressiveness and progression. Furthermore, our investigation reveals a compelling association between high VISTA expression, as quantified by H-score, and older patient age. This demographic subset exhibits a predisposition towards advanced disease stages, elevated Breslow index, and heightened incidence of ulceration compared to counterparts with diminished VISTA expression. High VISTA expression is also linked to a significant increase in inflammatory infiltration, suggesting an immunomodulatory role in the tumor microenvironment. Conversely, reduced VISTA H-score expression tends to be more frequently associated with pagetoid migration, indicative of a distinct tumor phenotype. The presence of positive VISTA expression in patients with uveal melanoma appears to correlate with certain pathological features. Specifically, among the cases examined, those with positive VISTA expression tended to have a higher stage (pT3a) compared to those with negative expression. Yet, due to the low number of cases and studies, further research with larger sample sizes is warranted to understand better the clinical implications of VISTA expression in uveal melanoma. All our findings illustrate the multifaceted interplay between VISTA expression levels and melanoma characteristics, underscoring its potential as a prognostic marker and therapeutic target.

Looking ahead, the prospect of future randomized studies holds promise in corroborating the robustness and clinical relevance of our findings. By further elucidating the intricate relationship between VISTA expression and melanoma progression, such endeavors may pave the way for enhanced diagnostic and therapeutic strategies, improving patient outcomes in this challenging neoplasm.

**Study limitations:** The current study included 105 cases, which might represent a limitation to our results due to the low number of patients. In the future, we hope to continue this research and expand our database.

## Figures and Tables

**Figure 1 ijms-25-04335-f001:**
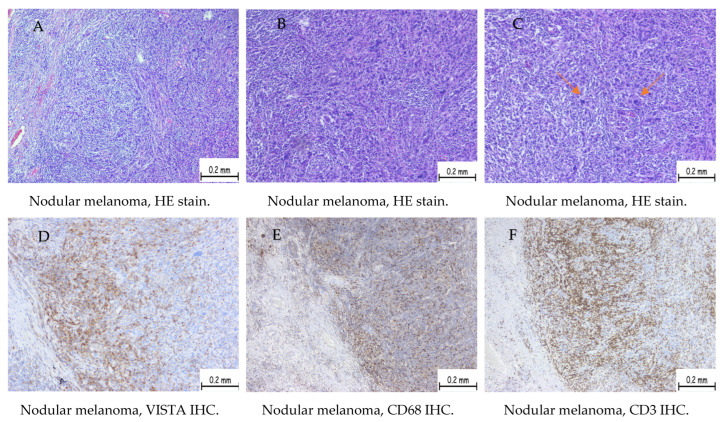
A case of nodular melanoma staged pT4b (more than 4 mm depth with ulceration of the surface epithelium) showing high VISTA expression (high H-score). (**A**) HE stain shows a solid tumor proliferation consisting of nests and plaques of tumoral cells showing epithelioid and fusiform shape, eosinophilic cytoplasm, and enlarged, pleomorphic nuclei. The brisk inflammatory infiltrate is present at the periphery of the tumor and diffusely spread between tumoral cells. (**B**) HE stain showing tumor proliferation consisting of the same changes described previously. Capture is taken in the center of the tumor. (**C**) HE stain showing tumor proliferation composed of highly pleomorphic cells. Mitoses are observed (arrows). Capture is taken in the center of the tumor. (**D**) Immunohistochemistry expression with VISTA showing a high number of cells with membrane and cytoplasmic staining at the periphery of the tumor. (**E**) Immunohistochemistry expression for CD68 showing membrane and cytoplasmic staining at the periphery of the tumor. (**F**) Immunohistochemistry expression for CD3 showing membrane staining at the periphery of the tumor. (**G**) Immunohistochemistry expression with VISTA showing a high number of cells with membrane and cytoplasmic staining in the central area of the tumor. (**H**) Immunohistochemistry expression for CD68 showing membrane and cytoplasmic staining in the central area of the tumor. (**I**) Immunohistochemistry expression for CD3 showing membrane staining in the central area of the tumor.

**Figure 2 ijms-25-04335-f002:**
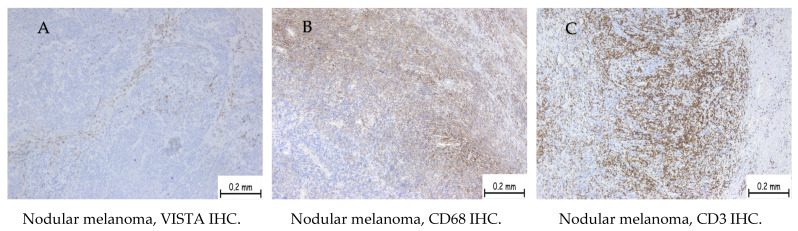
A case of nodular melanoma staged pT3a (more than 2 mm depth but less than 4 mm, without ulceration of the surface epithelium) showing low VISTA expression (low H-score). (**A**) Immunohistochemistry with VISTA showing membrane staining in a low number of cells. (**B**) Immunohistochemistry expression for CD68 showing membrane and cytoplasmic staining in a high number of cells. (**C**) Immunohistochemistry expression for CD3 showing membrane staining in a high number of cells.

**Figure 3 ijms-25-04335-f003:**
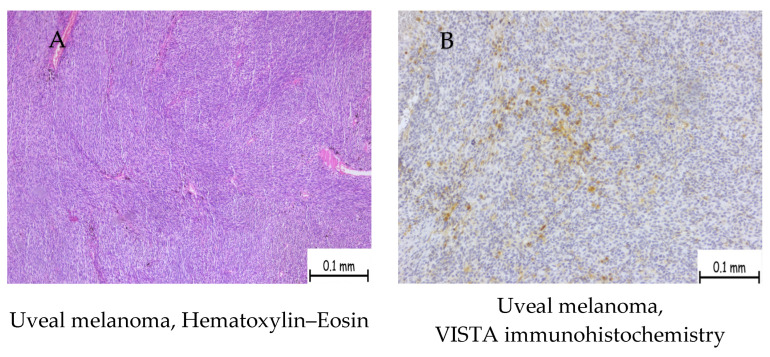
A case of uveal melanoma staged pT3a showing low VISTA expression (low H-score). (**A**) We observed a solid tumor proliferation consisting of nests and plaques of tumoral cells showing mostly epitheloid aspect, eosinophilic cytoplasm, and enlarged nuclei. (**B**) Immunohistochemistry expression with VISTA showing cells with membrane and cytoplasmic staining.

**Table 1 ijms-25-04335-t001:** Number of cases positive for VISTA in cutaneous melanoma.

Positive Cases	Nodular Melanoma	Superficial Spreading Melanoma	Uveal Melanoma
Total	56	9	8
High score	41	0	3
Low score	15	9	5

**Table 2 ijms-25-04335-t002:** Number of cases positive for VISTA in uveal melanoma.

No. Cases	Tumor Stage	Positive Cases	Vista High H-Score	Vista Low H-Score
3	pT1a	0	0	0
2	pT2a	1	1	0
7	pT3a	7	2	5

**Table 3 ijms-25-04335-t003:** Subtype of T cells identified in TME and their predominance.

	VISTA+	CD4+	CD8+
High score	41	10	31
Low score	15	5	10
Low score	9	9	0
Total	65	24	41

**Table 4 ijms-25-04335-t004:** Comparison between positive/negative VISTA expression in cutaneous melanoma.

Parameter	Positive VISTA Expression Patients *n* = 65		Negative VISTA Expression Patients *n* = 27		* *p* Value	OR	95% CI
	*n*	*%*	*n*	*%*			
**Age (years)**	64 (53–76)		60 (49–68)		0.11	-	-
**Advanced stage**	52	80	4	14.8	<0.001	23.0	6.6–66.9
**Ki-67 proliferation index** **≥ 25**	25	38.4	1	3.7	0.001	16.2	2.5–173.6
**Pagetoid spread**	21	32.3	6	22.2	0.4	1.6	0.6–4.8
**Clark levels 3 to 5**	58	89.2	20	74.0	0.1	2.9	0.9–8.8
**Inflammatory cells (moderate/high)**	59	90.7	1	3.7	<0.001	255.7	34.2–2647
**Breslow thickness > 2 mm**	52	80	4	14.8	<0.001	23.0	6.6–66.9
**Tumor ulceration**	36	55.3	7	25.9	0.01	3.5	1.2–8.8
**In situ component**	16	24.6	4	14.8	0.4	1.8	0.6–5.6
**Mitotic count (No)**	12 (5–20)		6 (3–8)		<0.001	-	-
**Positive surgical margins**	7	10.7	1	3.7	0.4	3.1	0.5–36.6
**Perineural invasion**	2	3.0	2	7.4	0.7	0.3	0.06–2.6
**Lymphovascular invasion**	8	12.3	1	3.7	0.3	3.6	0.4–41.9
**Microsatelitosis**	3	4.6	1	3.7	0.7	1.2	0.1–16.9
**Metastasis**	10	15.3	1	3.7	0.2	4.7	0.7–53.1

* Obtained from Fisher’s exact test for categorical variables, with Yates’s correction where expected frequencies of less than 5 were found/Mann–Whitney U-test for non-parametric continuous data; OR: odds ratio; CI: 95% confidence interval.

**Table 5 ijms-25-04335-t005:** Evaluation of populations with high/low levels of VISTA expression in cutaneous melanoma.

Parameter	High Levels VISTA Expression *n* = 42		Low Levels VISTA Expression *n* = 23		* *p* Value	OR	95% CI
	*n*	*%*	*n*	*%*			
**Age (years)**	72 (60–79)		57 (47–66)		<0.001	-	-
**Advanced stage**	41	97.6	11	47.8	<0.001	44.7	6.0–487.3
**Ki-67 proliferation index** **≥ 25**	19	45.2	6	26.0	0.18	2.3	0.7–7.4
**Pagetoid spread**	8	19.0	13	56.5	0.004	0.1	0.06–0.5
**Clark levels 3 to 5**	40	95.2	18	76.2	0.09	5.5	0.9–29.1
**Inflammatory cells (moderate/high)**	41	97.6	18	76.2	0.03	11.3	1.3–136.7
**Breslow thickness > 2 mm**	39	92.8	13	56.5	0.001	10.0	2.5–36.3
**Tumor ulceration**	30	71.4	6	26.0	<0.001	7.0	2.0–21.8
**In situ component**	7	16.6	9	39.1	0.06	0.3	0.09–1.0
**Mitotic count (No)**	14 (5–70)		9 (4–16)		0.05	-	-
**Positive surgical margins**	5	11.9	2	8.6	0.9	1.4	0.2–7.5
**Perineural invasion**	1	2.3	1	4.3	0.7	0.5	0.02–10.6
**Lymphovascular invasion**	6	14.2	2	8.6	0.7	1.7	0.3–9.0
**Microsatelitosis**	2	4.7	1	4.3	0.5	1.1	0.1–16.6
**Metastasis**	8	19.0	2	8.6	0.4	2.4	0.4–12.3

* Obtained from Fisher’s exact test for categorical variables, with Yates’s correction where expected frequencies of less than 5 were found/Mann–Whitney U-test for non-parametric continuous data; OR: odds ratio; CI: 95% confidence interval.

**Table 6 ijms-25-04335-t006:** Comparison analysis of patients based on high, low, and negative VISTA expression.

Parameter	High VISTA Expression Levels *n* = 42		Low VISTA Expression Levels *n* = 23		Negative Expression of VISTA *n* = 27		* *p* Value
	*n*	*%*	*n*	*%*	*n*	*%*	
**Age**	72 (60–79)		57 (47–66)		60 (49–68)		0.001
**Advanced stage**	41	97.6	11	47.8	4	14.8	<0.001
**Ki-67 proliferation index** **≥ 25**	19	45.2	6	26.0	1	3.7	<0.001
**Pagetoid spread**	8	19.0	13	56.5	6	22.2	0.004
**Clark levels 3 to 5**	40	95.2	18	76.2	20	74.0	0.03
**Inflammatory cells (moderate/high)**	41	97.6	18	76.2	1	3.7	<0.001
**Breslow thickness > 2 mm**	39	92.8	13	56.5	4	14.8	<0.001
**Tumor ulceration**	30	71.4	6	26.0	7	25.9	<0.001
**In situ component**	7	16.6	9	39.1	4	14.8	0.05
**Mitotic count (No)**	14 (5–70)		9 (4–16)		6 (3–8)		<0.001
**Positive surgical margins**	5	11.9	2	8.6	1	0.4	0.4
**Perineural invasion**	1	2.3	1	4.3	2	0.7	0.6
**Lymphovascular invasion**	6	14.2	2	8.6	1	0.3	0.3
**Microsatelitosis**	2	4.7	1	4.3	1	0.7	0.9
**Metastasis**	8	19.0	2	8.6	1	0.2	0.1

* Obtained from Chi-squared test for categorical data/Kruskal–Wallis one-way analysis of variance test for non-parametric continuous data.

## Data Availability

The data presented in this study are available on request from the corresponding author.

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
