# Peer review of "H-VISTA Immunohistochemistry Score Is Associated with Advanced Stages in Cutaneous and Ocular Melanoma"

_ijms, 2024, doi:10.3390/ijms25084335_

Round 1

Reviewer 1 Report

Comments and Suggestions for Authors

In the introduction the authors should report few sentences about the treatment of melanoma (Radiological and clinical findings in uveal melanoma treated by plaque interventional radiotherapy (brachytherapy): Visual atlas and literature review on response assessmentRadiological and clinical findings in uveal melanoma treated by plaque interventional radiotherapy (brachytherapy): Visual atlas and literature review on response assessment; Immunotherapy and radiotherapy in melanoma: a multidisciplinary comprehensive review; High-Dose-Rate Interstitial Brachytherapy (Interventional Radiotherapy) for Conjunctival Melanoma with Orbital Extension; Lancet. 2023 Aug 5;402(10400):485-502. doi: 10.1016/S0140-6736(23)00821-8. 

In the conclusion the authors could add that future randomized studies could serve to confirm the validity of this method

Author Response

Dear reviewer,

Thank you very much for your response. We sincerely appreciate your insightful suggestions. We have carefully considered them and are committed to implementing the necessary changes. Specifically, we will incorporate the suggested articles and provide concise descriptions of the outcomes of those aspects of the treatment. Furthermore, we will revise the conclusion to ensure it meets your expectations.

We will promptly make these adjustments and upload the revised manuscript according to your requirements. Thank you once again for your valuable feedback and guidance.

Best regards, 

Dr. Andreea Tinca

Reviewer 2 Report

Comments and Suggestions for Authors

Melanoma represents a public health issue. One of the biggest goals of current research is to develop new therapeutic options for patients affected by this aggressive tumor. In this research, the authors conducted a retrospective study including 105 patients diagnosed with cutaneous and ocular melanoma, stages varying from pT1a to pT4b, and we performed immunohistochemistry reactions with the new potential prognostic marker, VISTA (V-domain Ig suppressor of T cell activation). Immunohistochemical VISTA expression in cutaneous melanoma was associated with an advanced tumor stage; increased Breslow index; increased tumor proliferation index, ki67; and increased mitosis count. Over all evaluation of this manuscript gives two different impressions. Entire research sounds valid and the obtained results are interesting. However, this manuscript is not well prepared in well-understandable ways. I may suggest publication of this work in Int. J. Mol. Sci. However, some revisions are necessary to fix the latter part. Please see below.

1) For this work, approach and methods are important. Therefore, reading of experimental parts is important for better understanding. From this viewpoint, section order of Introduction => Materials and Methods => Results => Discussion => Conclusions is better than the current order.

2) Methods are not still well understandable. Addition of one figure to explain the used methods is recommended.

3) Although this work has practical importance, description of conclusion is so insufficient. Descriptions on research impacts and future perspectives in practical applications have to be added more.

Author Response

Dear reviewer,

Thank you so much for your insightful suggestions. We deeply appreciate your time and effort in reviewing our manuscript. We are fully committed to addressing each of your points.

Regarding the data, we will make the necessary changes to ensure accuracy and clarity. We understand your recommendation to alter the order in the first part, and we agree that it would be more appropriate given the context.

In the Materials and Methods section, we will try to provide a more comprehensive description to enhance understanding and reproducibility.

Lastly, we will devote attention to developing the conclusion in accordance with your expectations. We are optimistic that our revisions will meet your requirements.

Once again, we sincerely thank you for your valuable feedback and guidance.

Warm regards, 

Dr. Andreea Tinca

Reviewer 3 Report

Comments and Suggestions for Authors

1. Consider a more detailed statistical analysis to compare VISTA expression between cutaneous and ocular melanoma. Utilizing statistical tools like Kaplan-Meier survival curves could provide insights into how VISTA expression impacts overall survival and disease-free survival across these melanoma types.

2. A straightforward analysis could involve retrospectively comparing VISTA expression levels with the response to immunotherapy (e.g., PD-1/PD-L1 inhibitors) in the studied melanoma cases. This could be achieved by reviewing medical records for treatment outcomes and correlating these with VISTA immunohistochemistry scores.

3. To understand the broader immune landscape, perform immunohistochemistry for additional immune checkpoints (e.g., PD-L1, CTLA-4) on the same samples. Simple statistical correlation analyses between VISTA expression and these markers could reveal synergies or redundancies in the immune evasion mechanisms of melanoma.

4. Since VISTA expression was noted in inflammatory cells, further characterization of these cells using additional markers (e.g., for different lymphocyte subsets) could provide insights into the immune contexture of the tumor microenvironment. This could be achieved through dual immunohistochemistry staining or flow cytometry on available samples.

5. Implement digital image analysis for more objective and reproducible quantification of VISTA expression. This method can provide precise measurements of staining intensity and the proportion of positive cells, offering a more detailed understanding of VISTA's role in melanoma.

6. Expand the retrospective analysis to include patient demographics (age, sex, etc.) and clinical outcomes (recurrence, metastasis, survival time) in relation to VISTA expression levels. Simple statistical methods can be employed to identify any significant correlations.

7. If resources allow, conduct basic in vitro experiments to study the effect of VISTA blockade on melanoma cell lines. Observing changes in cell proliferation or cytokine production in response to VISTA-blocking antibodies could provide preliminary insights into potential therapeutic strategies.

8. A detailed examination of cases with negative VISTA expression could be insightful. Comparing these cases to those with high and low VISTA expression in terms of clinical outcomes and pathological features might uncover patterns that could guide further research or clinical approaches.

Author Response

Dear Editor,

We deeply appreciate your invaluable suggestions and the time you've dedicated to our study. It's important for us to address certain details regarding your requests. Regrettably, in our country, the development of the National Cancer Register is still underway, making it impossible for us to obtain patient data following the excision of the lesion. This limitation poses a significant challenge in our field, as it prevents us from collecting information on survival rates or metastatic occurrences.
Within our hospital, one of the major institutions in our region, we unfortunately got the chance to observe a few cases progressing from primary tumors to lymph node metastases. However, our ability to track these cases ends at this point. Given these constraints, and drawing upon data gathered from other studies, we have chosen to focus on implementing a feature with important clinical implications, accessible to clinicians in the future, benefiting patients.
This decision led us to concentrate solely on Immunohistochemistry (IHC), following a similar approach to the one taken with PD-L1, which is also diagnosed through IHC in melanoma. While PD-L1 is not currently our primary area of interest, it remains part of another study encompassing a broader range of cases. By developing this observational retrospective study for our specific cases, we aim to ascertain whether the results obtained could pave the way for the routine implementation of this method on a larger scale for all future diagnoses.

While Immunofluorescence (IF) is indeed an intriguing option, it does not align with our current focus due to the lack of clinical evidence in melanoma, as outlined by WHO/AJCC/ASCO-CAP guidelines. However, we are actively exploring other avenues, including the identification of inflammatory cell subtypes. To this end, we plan to conduct CD4/CD8 staining to ascertain the predominant cell populations (we will add it in the reviewed version). 

Thank you again for your time and patience, 

Sincerely,

Dr. Andreea Tinca

Round 2

Reviewer 2 Report

Comments and Suggestions for Authors

I recognized certain efforts by the authors. The manuscript becomes more readable. The vised version becomes acceptable.

Author Response

Dear Editor,

We thank you very much for your positive response! 

Reviewer 3 Report

Comments and Suggestions for Authors

Please answer the reviewer's comments one by one to ensure that each comment is taken seriously and answered.

Author Response

Dear Editor,

Thank you once again for dedicating your time to our manuscript. Following your feedback from the previous review, we have tried rephrasing and refining the article to better align with your requests. Additionally, we have sent it for proofreading and have attached the certificate for your reference.

We are fully open to your suggestions if you have any specific aspects you would like us to address or modify further.

Thank you sincerely for your continued guidance and support.

Best regards,

Dr. Andreea Tinca

Round 3

Reviewer 3 Report

Comments and Suggestions for Authors

I have no comments